# Evaluating MINFLUX experimental performance in silico

Zach Marin [1,2] & Jonas Ries [1,2,3] ✉

MINFLUX is a super-resolution microscopy technique with remarkable resolution for imaging and tracking. Its optical resolution is well understood under idealized conditions, but because of system complexity and experimental imperfections, actual performance can be much worse. Here we present SimuFLUX, a comprehensive and realistic simulator for MINFLUX. We use it to investigate performance limits caused by fluorophore dynamics, background, estimators, and misalignment, and to simulate experiments to optimize parameters and assess feasibility.

MINFLUX (minimal photon flux) microscopy measures the position of single fluorophores by targeted scanning of a patterned excitation beam featuring a minimum, reaching record precision for a given number of detected photons[1–3]. MINFLUX tracking has enabled novel biological insights into the mechanisms of motor proteins[4,5], and MINFLUX imaging has reached angström resolution[6,7]. Technical implementations are diverse[1,2,4,8–10], but all are highly complex in data acquisition and analysis. Nowadays, commercial MINFLUX microscopes are readily accessible to many researchers at advanced imaging facilities, yet obtaining high-quality data remains challenging because of the complexity of performing a measurement and because in non-ideal biological samples numerous imperfections can cause unforeseen artifacts[11,12]. How these experimental challenges impact MINFLUX quality and how they can be mitigated is still not well understood.

The theoretical limit of precision in MINFLUX has been studied extensively using the Cramér-Rao bound (CRB)[1,13,14] and Van Tree inequality calculations[15]. However, these do not include uncertainties or bias caused by imperfections stemming from optics (misalignment, aberrations), mechanics (speed of pattern scan and repositioning, vibrations), fluorophores (flickering, bleaching, movement, nearby fluorophores) or position estimators (which convert photon counts to approximate fluorophore positions) used in experiments. Simulations are needed to investigate how these imperfections influence the results, but available code can simulate only few of these confounders[1,13,14,16–20] (Supplementary information Table 1).

Here, we developed SimuFLUX, a comprehensive simulator for MINFLUX experiments. It includes realistic models of point spread functions (PSFs), fluorophores, microscope mechanics and estimators. It allowed us to answer long-standing questions on how imperfections affect MINFLUX accuracy, and we demonstrate how it can be used to optimize experimental parameters and to simulate entire experiments to judge feasibility of a project.

## Results

### SimuFLUX architecture

SimuFLUX performs a detailed and realistic simulation of a MINFLUX microscope (Fig. 1, Methods). The simulator scans a PSF over a sample in a defined pattern[1,2] or iterates through a set of PSFs[8] at a fixed position, measures the number of photons at each position/PSF, performs a position estimation, and then recenters the pattern by moving a fast scanner (e.g., an electro-optical deflector, EOD) and a slow scanner (a galvo[4,21] or a piezo[1,2]). The PSF model can be either experimentally calibrated[22,23] or calculated with a full vectorial PSF model that includes aberrations, misalignments, and detection through a pinhole. This allows us to simulate any MINFLUX approach, e.g. using 2D or 3D donuts or half-moon (bi-lobed) PSFs[8,24]. The fluorophore model simulates a collection of fluorophores that can switch on and off, show fast blinking (flickering) and bleaching, and can diffuse or move through defined trajectories (e.g, steps that describe moving kinesins[5] or oscillatory paths stemming from microscope vibrations). Available outputs (Fig. 1c) include the raw data, localizations in a format that can be imported into SMAP[25] for further analysis, statistics on performance, the CRB, as well as tracks and images.

SimuFLUX is provided as open source in both MATLAB and Python. It is designed to be easily used by non-experts and easily extended by developers and can be run locally or remotely on Google Colab. Experiments are either defined programmatically via MATLAB scripts or Jupyter Notebooks[26] using classes in the SimuFLUX library or via text files with a syntax that is an extension of the format used in Abberior MINFLUX sequence files. This allows non-expert users to

[1]Max Perutz Labs, Vienna Biocenter Campus (VBC), Vienna, Austria. [2]University of Vienna, Center for Molecular Biology, Department of Structural and Computational Biology, Vienna, Austria. [3]University of Vienna, Faculty of Physics, Vienna, Austria. ✉e-mail: jonas.ries@univie.ac.at

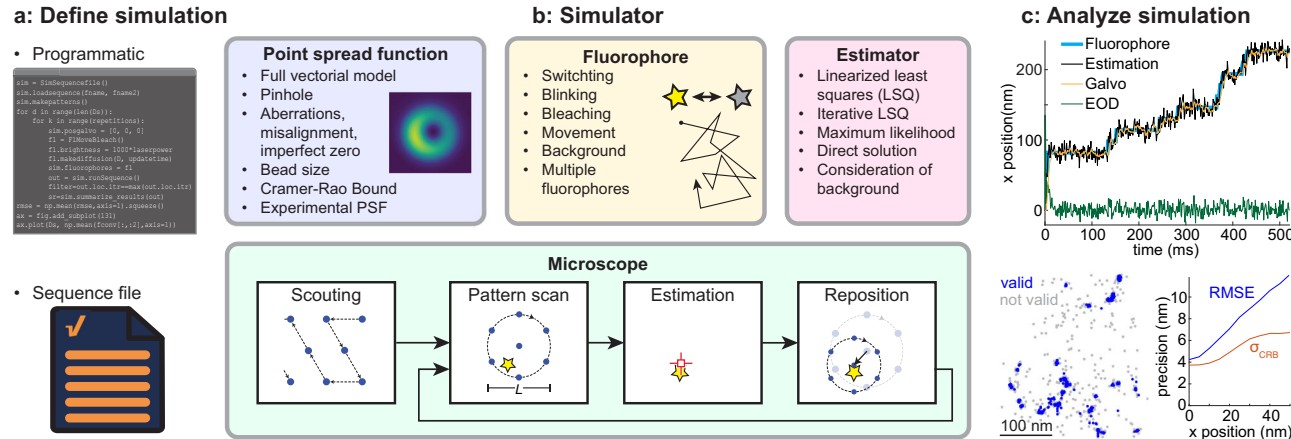

**Fig. 1 | SimuFLUX overview. a** Simulations are defined programmatically or via sequence files. **b** The simulator performs comprehensive simulations of a MINFLUX experiment using realistic models for the point-spread function, fluorophores, and microscope optics/software/mechanics. LSQ: least squares. **c** The output comprises raw data, simulated images, and statistics on performance. EOD: electro optic deflector, RMSE: root mean squared error, CRB: Cramér-Rao Bound.

directly simulate an actual Abberior MINFLUX experiment with precise parameters, with the caveat that we replaced some propriety algorithms (e.g., the estimator), by published algorithms. All simulations in this manuscript are available as example scripts, which also serve as tutorials.

We now illustrate with several examples how SimuFLUX allows us to understand fundamental performance limitations of MINFLUX under realistic conditions, and how it helps us to develop improved MINFLUX approaches to overcome these limitations.

## Aberrations, vibrations and background

Careful alignment of MINFLUX optics, particularly of the phase pattern, is known to be of importance[16] and indeed we find that shifting the phase pattern by a few 100 micrometers off the optical axis (Fig. 2a, Supplementary information Fig. 1) results in a z-dependent bias and additional uncertainty (i.e., standard deviation) in position estimates. Note that the CRB does not reflect this deteriorated performance, highlighting the need for realistic simulations. On the other hand, a misaligned pinhole has a negligible effect on performance apart from reducing the photon detection efficiency (Supplementary information Fig. 2). Most aberrations and a background have a much stronger impact when using a 3D donut compared to a 2D donut or half-moon PSFs (Fig. 2b, Supplementary information Fig. 3). Vibrations during the measurement lead to additional uncertainties on the order of the vibration amplitude and show little frequency dependence (Supplementary information Fig. 4).

A fluorescence background from nearby fluorophores, autofluorescence, or an imperfect zero of the PSF is ultimately the limiting factor for MINFLUX precision[3,11,20]. We investigated the impact of these sources of background (Supplementary information Fig. 5) and found that they not only deteriorate the localization precision but also cause a strong bias of the estimator. This prompted us to develop better estimators.

Estimators must be simple and fast when run on a field-programmable gate array (FPGA) to allow for online position feedback on a sub-millisecond time scale[3]. We developed a new iterative estimator, and for 1D localizations a direct closed form estimator, which greatly extend the field of view over which there is low bias as compared to the standard first-order estimator (Fig. 2c, Supplementary information Fig. 6). These estimators maintain mathematical simplicity[1,20], which suggests they could be implemented efficiently on an FPGA. We found that bias arising from background disappears if the background can be estimated precisely and is subtracted from photon counts or is used as a free fitting parameter in the estimator (Fig. 2d). An estimator bias also helped us explain a phenomenon the user base experiences with the Abberior MINFLUX: often, the estimated position creeps slowly towards the true position during the first iterations, with an error of tens of nanometers. Simulations could recapitulate this behavior when the first coarse localization step was inaccurate, when beads are used that–due to their size–reduce the zero contrast, or when background estimation is imprecise (Supplementary information Fig. 7).

## Fluorophore kinetics

MINFLUX assumes constant brightness during probing of positions, but fluorophores bleach and can show flickering on many time scales[27,28]. We show that flickering leads to an additional error on the order of up to 10 nm, which can be mitigated by repeated scanning of the coordinate pattern within the same pattern scan time (Fig. 2e, Supplementary information Fig. 8). Interestingly, the additional error $\sigma_{\text{fl}}(r)$ does not depend on the photon counts, but only on the dye kinetics. The localization precision can be calculated as $\text{STD}(r, N)^2 = \sigma_{\text{fl}}(r)^2 + \sigma_{\text{CRB}}(N)^2$, where $\sigma_{\text{CRB}}$ is the standard deviation for $N$ photons without flickering. Depending on the dye kinetics, a high number of repetitions $r$ may be needed (>20) to sufficiently reduce the error, which is not feasible with current technologies. Thus, strongly flickering fluorophores such as quantum dots should be used with care. Our results show the same trend as related studies[19,29].

Next, we simulated realistic experiments for MINFLUX imaging and tracking using the default Abberior sequences.

DNA-PAINT has become increasingly popular for MINFLUX imaging because of the high photon counts detected from single fluorophores and the simplicity of sequential multi-color measurements, but the diffusing imaging strands (density ~1/μm³) cause a background that varies on the spatial and temporal scale of the pattern scan, potentially causing artifacts. Indeed, our simulations on the example of a nuclear pore complex[30] (Fig. 2f) showed a reduced localization accuracy and many invalid localizations. Performing identical simulations with invisible imaging strands mitigated that problem (Fig. 2g). Thus, DNA-PAINT with quenched imaging strands[31] might be preferable for MINFLUX, provided they do not lead to additional flickering, which remains to be investigated.

MINFLUX seems especially suited for fluorophores with a low photon budget, such as photo-activatable/photo-convertible fluorescent proteins (PAFP, as in PALM[32]). Unfortunately, in our simulations many PAFPs bleached during the scouting phase and first iterations,

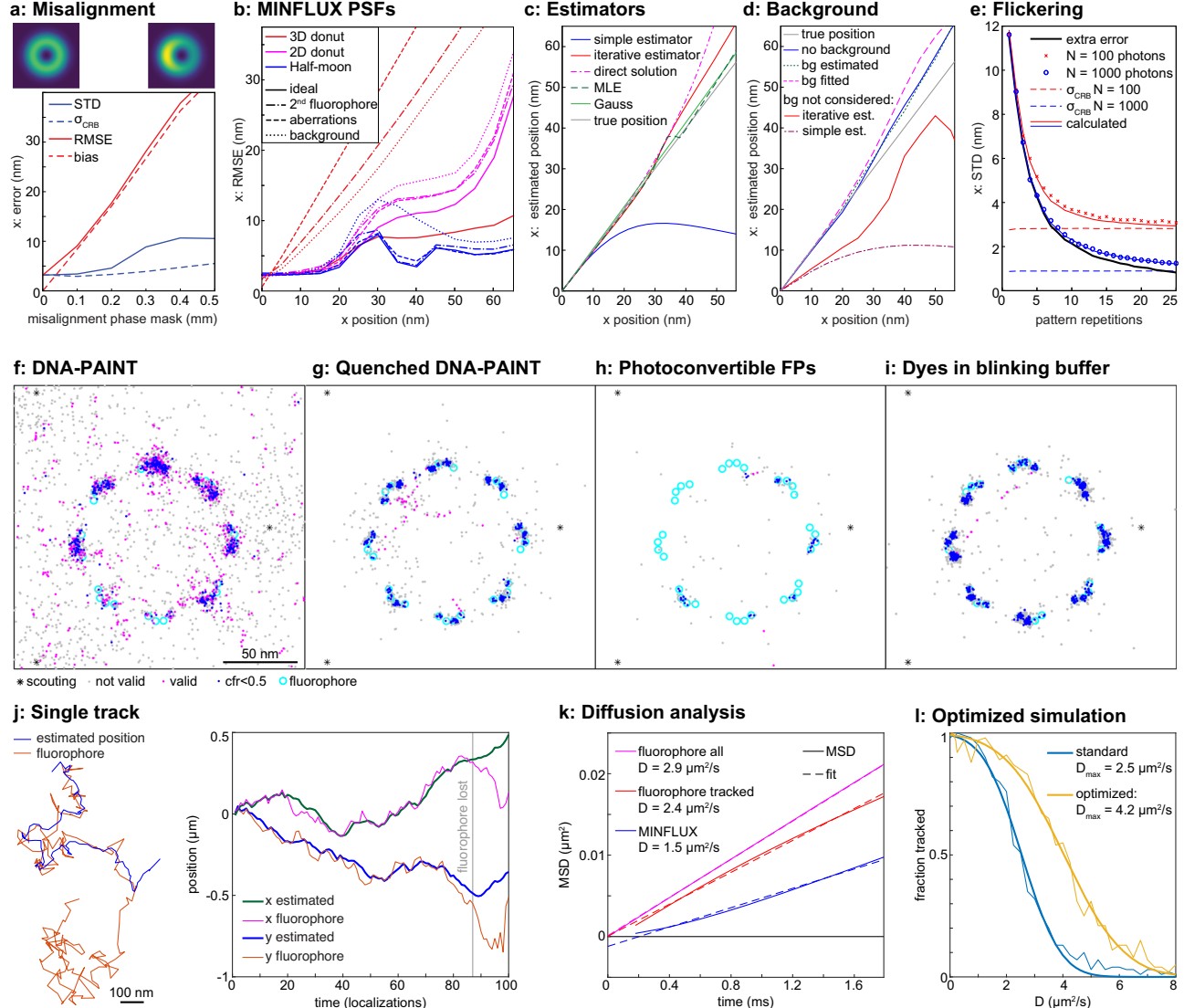

**Fig. 2 | Simulations of MINFLUX performance in realistic experimental conditions. a** Misalignment of the phase mask results in a bias and increased error in position estimates (see Supplementary information Fig. 1 for more details). Shown are PSFs for no misalignment and a misalignment of 0.5 mm in the sample x-direction. STD: standard deviation, RMSE: root mean square error, $\sigma_{CRB}$: localization precision from the Cramér-Rao Bound (CRB). **b** Comparison of MINFLUX PSFs. RMSE in dependence of the field of view (x position of the fluorophore) for 2D and 3D donuts and half-moon PSFs from PhaseFLUX[8], simulated for the ideal case and in presence of background, in presence of a nearby fluorophore and with aberrations. Example PSFs are shown in Supplementary information Fig. 3. **c** Estimators (see Methods for equations) comparing the standard linearized least squares (LSQ) with an iterative LSQ estimator, a direct fit of a quadratic model, and Maximum Likelihood Estimation (MLE). The Gaussian PSF was fit with a simple estimator. See "Methods" and Supplementary information Fig. 6 for more details. 500 photons/localization. **d** Background bias for the iterative and conventional LSQ estimator and mitigation via subtracting background from photon counts and via

fitting the background. 200 signal photons, 200 background photons at x = 0. **e** Fluorophore flickering. Averaging over repeated pattern scans mitigates loss in precision. $t_{on} = t_{off} = 100$ μs. **f–i** Comparison of different imaging modalities on simulated nuclear pore complexes. MINFLUX using DNA-PAINT with standard (**f**) and fluorogenic (**g**, quenched) imaging strands, using photoconvertible fluorescent proteins (**h**, as in PALM), and organic dyes in a blinking buffer (**i**, as in dSTORM). **j–l** In silico optimization of MINFLUX tracking. **j** Example track of a molecule diffusing in 2D with $D = 3$ μm²/s. When the jump of a molecule is too large (see Supplementary information Fig. 10a), the microscope cannot follow, and the fluorophore is lost. **k** The limited time resolution in combination with loss of fast dynamics leads to an underestimation of the diffusion coefficient in MINFLUX and non-linear mean square displacement (MSD) over time. **l** A systematic optimization of laser power, pattern size, photon limit and pattern dwell time allowed us to substantially increase the time resolution. 90 consecutive localizations were considered a successful track, $D_{max}$ denotes the diffusion coefficient for which half of the tracks were successful.

leading to a reduced effective labeling efficiency (Fig. 2h). A simulation of organic dyes in a thiol-containing blinking buffer (as in dSTORM[33,34]) lead to a higher effective labeling efficiency and better resolution (Fig. 2i).

In all these approaches, the density of activated fluorophores must be tuned carefully. At high densities, nearby active fluorophores outshine the target fluorophore close to the PSF zero and thus lead to a strong localization error (Supplementary information Fig. 5g,h). Low

densities further increase the already long imaging times. We thus optimized the activation density in silico (Supplementary information Fig. 9), as a basis for experiments.

## Tracking diffusion
MINFLUX is especially advantageous for single fluorophore tracking, where it greatly increases the spatial and temporal resolution as well as the track length when compared to camera-based tracking[3]. However,

it is difficult to track fast fluorophores, e.g., membrane diffusion with a diffusion coefficient of $D > 1 \, \mu m^2/s$. We explored the cause of this limitation by systematically investigating the impact of scan pattern shape and size, laser powers, photon limits, pattern dwell times and detection dead times on the maximal diffusion speed that can be tracked (Fig. 2 j-l, Supplementary information Fig. 10, Supplementary information Table 2). We identified microscope dead times, which occur during scanner movement or during estimator calculations on the FPGA, as a main limitation and found that optimization of pattern size, photon limit and pattern dwell time increased the measurable diffusion coefficient by 70% over that captured by the standard sequence. These observations are corroborated by a recent experimental paper, which also found that reducing photon limit and pattern dwell time increased the measurable diffusion coefficient estimation until hardware timing limits were reached[35]. Of note, insufficient time resolution caused an underestimation of the diffusion coefficient, thus MINFLUX diffusion measurements should be analyzed with caution.

## Discussion

SimuFLUX will be particularly useful for three directions of applications. First, it will be of use for developers, as it allows testing and optimizing new MINFLUX approaches and algorithms in silico. New probe design ideas could be optimized for MINFLUX, and new estimators can be prototyped to extend field of view or improve the localization precision. Second, it will continue to improve our theoretical understanding of MINFLUX and its dependence on various conditions. Third, we expect it to be an invaluable tool for all MINFLUX users as it will allow them to improve data quality by computationally optimizing the numerous experimental parameters available on commercial systems, and to use realistic simulations to judge the suitability of MINFLUX to answer their biological questions before the start of a project.

## Methods

### Software practices

This software is open source, released under the GNU General Public License v3.0. We welcome contributions from the community.

Complete versions of this simulator are written in both MATLAB and Python. While MATLAB is by default easy to install and use, inconsistencies in Python package management can often make dissemination of software difficult. To ensure the software is easy to install, we minimized the dependencies of the Python library. Only numpy[36] and scipy[37] are required to run the software, and IPython/Jupyter notebooks[26] are needed to interact with the software via the example notebooks. We also published the Python notebooks on Google Colab to allow researchers to use the simulator without installing anything.

The software is written using an object-oriented programming paradigm, with parts of the microscope and sample (described below) implemented as distinct classes for clear, logical separation of behavior and tasks. Because of this application programming interface, this simulator could be incorporated into popular third party software such as napari[38].

### Microscope model

The microscope is implemented as a point-scanning system with a descanned large range 'slow' scanner (a galvo[4,21] or a piezo[1,2] for x-y scanning and optionally a deformable mirror for z-scanning[21]) and optionally a 'fast' scanner (e.g., an electro-optical deflector, EOD), which only affects the excitation path, i.e., it is not descanned in detection, in line with current implementations[1,2,4,8–10]. Excitation positions and timings of these scanners can be controlled from the "posgalvo" and "posEOD" properties of the "Simulator" class. The excitation profile is defined by a point spread function (PSF), or a collection of PSFs, which are implemented as children of the "Psf" class

(see PSF model for details). Photon collection is assumed to be proportional to the PSF excitation intensity multiplied by the PSF detection efficiency, with the pinhole centered onto the pattern center, as the fast scanner is not descanned in the detection arm. The collected intensity is governed by a "Fluorophore" class or child "intensity" function. Aberrations and optical misalignment are modeled in the PSF. System vibrations can be modeled using moving fluorophores ("FlMoving" class or children).

### Experiment model

MINFLUX experiments are established in silico via the "Simulator" class, which takes a microscope model, a single fluorophore or a collection of fluorophores, one or more patterns, an estimator, and feedback on the scan position. Patterns are generated as a series of PSF positions, using the "definePattern" function in the "Simulator" class or a collection of PSFs. Sequences are lists of "Simulator" patterns and components (e.g. estimators) that can tell the simulator to run patterns, collect signal, use an estimator to evaluate fluorophore positions, update the scanner positions, and repeat.

In the "Simulator" class, each fluorophore position is probed with the full sequence provided to the simulator.

### Abberior MINFLUX model

The "SimSequencefile" class extends the "Simulator" class to allow users to load Abberior sequence files and simulate their behavior. These sequence files can be adapted to simulate any other MINFLUX microscopes.

Abberior sequences files contain a list of MINFLUX scanning iterations and their properties, as well as global parameters shared across all iterations. Because Abberior estimators are proprietary, we also include a custom JSON file (defaultestimators.json) that complements Abberior sequences files and indicates which estimators are used after each Abberior iteration. This file pulls from estimators implemented in as functions in the "estimators" folder. Once files are loaded, patterns (from "definePattern", described above) are generated for each iteration.

For each simulated experiment, first, a scouting pattern formed by moving the excitation PSF over the field of view in a hexagonal grid is used to identify candidate positions of fluorophores (scouting in Fig. 2f-i and Supplementary information Fig. 8 legends). This model uses a Gaussian excitation PSF to search for fluorophores. This is an approximation of the Abberior method of scanning a pinhole around the rim of a fixed excitation donut PSF (pinhole orbit)[39]. We replicate this by scanning a Gaussian in a hexagonal pattern around a set point. Note that this does not consider bleaching in the donut outside the detection PSF, thus we underestimate bleaching during scouting. Any position eliciting signal is marked as a candidate. Each candidate fluorophore is probed until the photon limit ("phtLimit") is reached, until it is bleached or lost (collected photons below "bgcThreshold") or, in case of probing the pattern center, until it fails the CFR check. The CFR is the ratio of the signal collected when illuminating a point at the centroid of a scanning pattern to the average signal of the rest of the pattern. CFR check fails if the CFR value is greater than the parameter "ccrLimit". Both CFR check and brightness check can fail for the number of times defined by the "stickiness" parameter. Localizations that pass the photon threshold or CFR checks are marked as valid, while those that do not pass either of these checks are marked as invalid (e.g., in Fig. 2f-i and Supplementary information Fig. 9).

Additional filtering of the CFR value allows for rejecting poor localizations (e.g. cfr<0.5 in Fig. 2f-i and Supplementary information Fig. 9), for example caused by nearby fluorophores or a large offset between fluorophore and pattern center.

Microscope dead times arising from hardware movement and position estimator calculations were measured on a commercial Abberior MINFLUX with iMSPECTOR version 16.3.21317-m2410-win64-

Minflux. EOD movement took 0.011 ms, estimator calculations took 0.015 ms, and galvo movement took 0.04 ms. A recent paper[35] that measured these times with a previous firmware version on a different MINFLUX microscope found EOD movement takes 0.005 ms and galvo movement takes 0.02 ms. For simulations in this paper, we assumed the newer firmware.

The "SimSequencefile" class currently understands the Abberior iteration specific parameters "ccrLimit", "pattern", "patDwellTime", "patRepeat", "pwrFactor", "patRepeat", "maxOffTime", "phtLimit", and "wavelength", and the global parameters "bgcThreshold", "ctrDwell-Factor", "damping", "headstart", "id", "loclimit", "stickiness", and "fieldGeoFactor" from the "field" structure. These parameters are described in detail in an annotated YAML file (Tracking_2D.yaml, uploaded to GitHub, see Code Availability section) derived from an Abberior JSON sequence file. The "algo" setting from the "field" structure currently only performs scouting for fluorophores by scanning a Gaussian beam in a hexagonal grid ("hexgrid") pattern, but this could be easily extended to other search patterns in the "makescoutingpattern" class function.

## PSF model

PSFs models are implemented as derivates of the "Psf" class, with each child class containing different methods to express PSF shape. PSF profiles can be expressed analytically, as in the "PsfGauss2D" and "PsfDonut2D" classes, or via a vectorial PSF model ("PsfVectorial" class) based on Leutenegger et al.[24]. The "PsfVectorial" class calculates the electromagnetic field using fast Fourier transforms by taking the vectorial Debye diffraction integral. For the different beam shapes, the phase patterns of the input beam at the back focal plane can be modified via the "phasepattern" argument to the "calculatePSFs" function in the "PsfVectoral" class, simulating use of e.g. a spatial light modulator. We additionally introduced a pinhole (controlled via the "setpinhole" function), which can be misaligned via a shift (see Supplementary information Fig. 2). Multiplication of the excitation PSF at a specific pattern position by a detection PSF through a pinhole at the pattern center is used to simulate the effects of both the excitation and detection PSFs in the optical system. We also introduce a bead size parameter (controlled via the "beadsize" function) that, when set to greater than zero, will result in a bead being convolved with the PSF. Large beads introduce an apparent background into the PSF shape by averaging out the zero, which can be useful for simulating the source of tracking errors (see Supplementary information Fig. 7).

Default parameters for the vectorial PSF—including, but not limited to, magnification, numerical aperture, refractive index of the immersion solution and the sample, pixel size, and aperture diameter—can be found, with descriptions, in the "default_microscope.yaml" file included with the software. This file is loaded, and its parameters are applied by default when generating a vectorial PSF.

Finally, experimentally calibrated PSFs[22,23] can be directly loaded ("PsfVectorial" class), combined with pinhole detection, and used for simulations.

## Fluorophore model

Individual fluorophores are implemented as children of the "Fluorophore" class, with changes to position ("position" function) and intensity ("intensity" function) modeled as needed in, e.g., fluorophores that move ("FlMoving" class) and those that bleach ("FlBleach" class). Collections of fluorophores, which can model everything from multiple diffusing particles to fixed nuclear pore complexes, are implemented using the "FlCollection" class and its children.

Moving fluorophores provide the coordinates at defined time points $t_i$ via the "position" function. Positions can be defined as mathematical functions or by directly providing the position $x_i(t_i)$ on a sufficiently dense time axis. Note that currently each point

measurement is performed at a fixed fluorophore position; thus the time resolution $dt$ should be below the point detection time. Finer sampling can be implemented by probing each point several consecutive times.

Tracks for specific motions can be pre-calculated.

Diffusion is modeled by generating jump lengths $W_k$ for each coordinate $k$ from a normal distribution:

$$W_k = \sqrt{2D\,dt} \cdot \mathrm{randn}()\qquad(1)$$

where $dt$ is the simulated time window length and $D$ ($\mu m^2$/s) is the diffusion coefficient. The position of a fluorophore at time $t_i$ is the cumulative sum of the jumps that occurred until this time point.

Stepping was modeled by generating the time interval between steps from an exponential distribution with a mean equal to the step dwell time $t_{step}$. Steps were then sampled with the simulated time window length $dt$, and the step position at time $t_i$ was the cumulative sum of the steps that occurred until this time point.

Fluorophores in "FlBleach" were initialized with a total number of photons set to a random number taken from an exponential distribution with a mean equal to a specified mean number of photons. To model bleaching, the number of detected photons in a measurement was subtracted from this total until it reduced to zero. Note that we subtracted the number of photons before application of the pinhole, as these correspond to the emitted photons.

Blinking was modeled in "FlBlinkBleach" by using random exponential distributions for the on and off times of the fluorophore, with means $t_{on}$ and $t_{off}$, respectively, to pre-calculate the emission state over time as the time of on- and off-switching events. The "measure" function calculates the fraction of the measurement time window during which the fluorophore was in its on-state and this "measure" function is called by the "intensity" function to calculate the total number of fluorophores emitted in a time window.

Photoconvertible fluorescent proteins, PAINT probes, and dyes in blinking buffers were modeled by varying the photon budget, $t_{on}$ and $t_{off}$, and number of fluorophore reactivations (return from a long-lived dark state, as opposed to bleaching). Photon budgets are approximated based on average experimental value observed in our lab. General trends for comparing photon budgets, such as that fluorescent proteins are dimmer than organic dyes[40], are followed. $t_{on}$ and $t_{off}$ can be gathered from the literature[41,42], but simulations in this paper use order-of-magnitude approximations of these values. Number of fluorophore re-activations ("activations" property in the "FlCollectionBlinking" class) are used to simulate differences between PAINT (infinite reactivations), STORM/dSTORM (2 reactivations on average[42]) and PALM (no reactivations[41]).

## Estimators

Estimators are functions that take positions of an excitation PSF and the fluorophore intensity response as inputs and return an estimation of the fluorophore's position. Here, we consider simple estimators that can potentially be deployed on an FPGA for real-time feedback on the scanning position. Estimators can be evaluated as part of the simulator's sequence by defining them as a "Simulator" class component via its "defineComponent" function.

Here, we follow Balzarotti et al.[1]. Typically, we perform $K$ measurements with the PSF $f(\mathbf{x})$ positioned at $\mathbf{b}_i$ for $i \in K$. For 2D or 3D measurements $\mathbf{b}_i$ is a vector. Then the expected number of detected photons in each position is

$$I_i = I_0\big(f(\mathbf{x} - \mathbf{b}_i) + bg\big).\qquad(2)$$

where $I_0$ describes fluorophore brightness and laser intensity, and $bg$ denotes a fluorescent background. The probability of measuring a

fluorophore with the illumination profile is defined as

$$p_i = \frac{I_i}{\sum_{j=1}^{K} I_j}. \tag{3}$$

Note that any PSF normalization factor, as well as $I_0$, cancel out.

The estimation task consists of estimating $x$ from experimentally measured photon numbers $\hat{n}_i$ with $\hat{N} = \sum \hat{n}_i$ being the total number of detected photons, or from the measured probabilities $\hat{p}_i = \hat{n}_i/\hat{N}$.

In the following, we will review common estimators from the literature[1] and derive estimators (see the Mathematica notebook SimuFLUX_estimators.nb in the Supplementary Software).

### Linearized least squares estimator

The least squares (LSQ) estimator minimizes

$$S = \sum_i (\hat{n}_i - I_i)^2 = \sum_i (\hat{p}_i - p_i)^2. \tag{4}$$

Note that we can use the background-free case, if we subtract the background from $\hat{n}_i$:

$$S = \sum_i \left( (\hat{n}_i - I_0 bg) - I_0 f(x - b_i) \right)^2 \tag{5}$$

Thus, in the following, we will not consider the background. A common approach to solving an LSQ problem is to linearize $p(x)$. Linearization around $x = 0$ results in the simple estimator (see Balzarotti et al[1]., Eq. S43-S48):

$$x_{\text{est}} = \left( J^T J \right)^{-1} J^T (\hat{p} - p(0)), \tag{6}$$

where $J = \partial p_i / \partial x_j$ is the Jacobian matrix.

The PSF (integral normalized to $A$) of a MINFLUX donut can be approximated as[1]:

$$f_d(x) = A \frac{x^2}{4\pi \sigma^4} e^{-\frac{x^2}{2\sigma^2}}, \tag{7}$$

With $L_\sigma^2 = L^2/8\sigma^2$, the linearized least squares estimator for the donut is[1]:

$$x_{\text{est}} = \frac{1}{L_\sigma^2 - 1} \sum_i \hat{p}_i b_i. \tag{8}$$

Derived for $K = 3$ measurement positions in an orbit of diameter $L$, this estimator is also a good estimator for more measurement points. Because of the expansion around $x = 0$ it becomes biased when the fluorophore is not at the center of the scan pattern. Balzarotti et al. debiased this estimator, however, and this only works in a relatively small range around $x = 0$.

The center of the donut can be approximated by a quadratic function (first order approximation of Eq. 7):

$$f_q(x) = A \frac{x^2}{4\pi\sigma^4}. \tag{9}$$

where $\sigma$ describes the steepness of the PSF. As can be seen in Supplementary information Fig. 6, this approximation is very good in the range of interest, i.e., when $|x| < L/2$. The linearized LSQ solution for the quadratic PSF is:

$$x_{\text{est}} = -\sum_i \hat{p}_i b_i. \tag{10}$$

For a Gaussian PSF,

$$f_q(x) = A e^{-\frac{x^2}{2\sigma^2}}, \tag{11}$$

the linearized LSQ estimator around $x = 0$ is (see Simu-FLUX_estimators.nb):

$$x_{\text{est}} = \frac{1}{L_\sigma^2} \sum_i \hat{p}_i b_i \tag{12}$$

if the center is not measured, and

$$x_{\text{est}} = \left( K_o + e^{L_\sigma^2} \right) \frac{1}{K_o L_\sigma^2} \sum_i \hat{p}_i b_i \tag{13}$$

if the center is measured ($K_o$ is the number of points in the circular orbit).

### Iterative least squares estimator

To extend the range of the LSQ estimators, we can linearize $p_i$ around an arbitrary location $x_0$ to calculate the Jacobian $J_{x0}$ at this location and use $x_0$ and $J_{x0}$ in Eq. 6.

In an iterative approach, starting with $x_0 = 0$, in every iteration we can set $x_{0,i} = x_{\text{est}, i-1}$. This estimator converges within a few iterations (Supplementary information Fig. 6). In the Mathematica notebook (SimuFLUX_estimators.nb) in the Supplementary Software, we calculated the Jacobian $J_{x0}$ analytically for $K_o = 3$, $K_o = 4$ and $K_o = 6$ probing positions at the orbit, with and without probing the pattern center.

### Other estimators for 1D localization

For the 1D case, we can derive simple analytical estimators either as a maximum likelihood estimator (MLE) or using a direct solution of a system of equations by equating measurements with the theoretical probabilities and solving for the position. Both approaches are relevant also for 2D or 3D MINFLUX measurements, as often two measurements on opposing sides of the orbit, with or without center measurement, can be treated as a 1D measurement. Also, implementations like the interferometric MINFLUX[4] or PhaseFLUX[8] consist of sequential 1D measurements for each coordinate.

The maximum likelihood estimator (MLE) for the quadratic PSF with probing points at $b_i = [-L/2, L/2]$ has been derived by Balzarotti et al[1].:

$$x_{\text{est, 1}} = \frac{L}{2} \frac{\sqrt{\hat{n}_1} - \sqrt{\hat{n}_2}}{\sqrt{\hat{n}_1} + \sqrt{\hat{n}_2}} , \quad x_{\text{est, 2}} = \frac{L}{2} \frac{\sqrt{\hat{n}_1} + \sqrt{\hat{n}_2}}{\sqrt{\hat{n}_1} - \sqrt{\hat{n}_2}}. \tag{14}$$

A center measurement $\hat{n}_0$ can be used to calculate the likelihood for both solutions to choose the correct one.

For $K = 2$ we can directly solve the equations

$$\hat{p}_i = = p(b_i) \tag{15}$$

for $x$ and $I_0$. Interestingly, for the quadratic approximation, this solution is identical to the MLE solution (Eq. 14, see SimuFLUX_estimators.nb).

For $K = 3$, i.e., three measurements $\hat{n}_0, \hat{n}_1, \hat{n}_2$ at $b_i = [0, -L/2, L/2]$, we can solve for $x$, $I_0$ and the background $bg$, i.e., treat the background as a free fitting parameter, instead of calibrating it with a separate measurement. Then, the estimated position is[20],

$$x_{est} = \frac{L}{4} \frac{\hat{n}_2 - \hat{n}_1}{2\hat{n}_0 - \hat{n}_1 - \hat{n}_2}. \tag{16}$$

## Data analysis

Cramér-Rao bounds (CRBs) for each PSF were calculated numerically following the framework in Masullo et al[13].: the Fisher information matrix was computed from the derivatives of the normalized PSF patterns and inverted to compute the lower bound. To get the Fisher information matrix, we first evaluated the pattern scan for each fluorophore at position $x - \varepsilon/2$ and $x + \varepsilon/2$ and then computed the numerical derivative for each axis. We then use the product of these derivatives pairwise to compute the Fisher information matrix such that it contains all information from the simulation, including background, the presence of other fluorophores, and aberrations in the scan PSF.

In addition to using the CRB, the success of experiments was evaluated on the following metrics.

The root mean squared error (RMSE) was defined as

$$\text{RMSE} = \frac{1}{Q}\sum_{i=1}^{Q}(\hat{x}_i - x_i)^2 \qquad (17)$$

where $Q$ was the total number of fluorophores in the sample, $\hat{x}_i$ was the estimated position of the fluorophore and $x_i$ was the true position of the fluorophore.

The standard deviation (STD) was defined as

$$\text{STD} = \sqrt{\frac{1}{Q}\sum_{i=1}^{Q}\left|(\hat{x}_i - x_i) - \bar{x}\right|^2} \qquad (18)$$

where $Q$ was the total number of fluorophores in the sample, $\hat{x}_i$ was estimated position of the fluorophore, $x_i$ was the true position of the fluorophore, and $\bar{x} = \frac{1}{Q}\sum_{i=1}^{Q}(\hat{x}_i - x_i)$.

The bias was defined as

$$\text{bias} = \bar{x} = \frac{1}{Q}\sum_{i=1}^{Q}(\hat{x}_i - x_i) \qquad (19)$$

where $Q$ was the total number of fluorophores in the sample, $\hat{x}_i$ was estimated position of the fluorophore, and $x_i$ was the true position of the fluorophore.

## Experimental single-molecule data

Single-molecule tracks (Supplementary information Fig. 7) were acquired as described previously[43]. In short, Atto647N coupled to single-stranded DNA was applied at low concentrations on a coverslip and then dried. We used an Abberior MINFLUX instrument with a 2D tracking sequence to acquire the data. Sequence settings for the last iteration: L: 75 nm; patRepeat: 3; patDwellTime: 0.3 ms; phtLimit: 500; stickiness: 4; ccrLimit: 0.9; laser power: 280 μW.

## Data availability

All data is generated by the simulation code, which is available on GitHub as described in the code availability statement.

## Code availability

Source code is available under a GPLv3 license at https://github.com/ries-lab/SimuFLUX/. A version of record is available at https://doi.org/10.5281/zenodo.17523755[44]. Example workflows/tutorials are provided with this code as MATLAB scripts and as Jupyter notebooks or can be opened on Google Colab without any installation of software: https://github.com/ries-lab/SimuFLUX?tab=readme-ov-file#google-colab.

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

## Acknowledgements

We thank Nikolay Sergeev, Sha Hao and Dr. Ioannis Pitsios for testing this software. We thank Dr. Sarah Schweighofer, Nikolay Sergeev, Dr. Takahiro Deguchi, Dr. Ioannis Pitsios, and Dr. Francesco Reina for manuscript feedback. We thank Dr. Takahiro Deguchi for experimental MNFLUX tracks and Dr. Fabian Hauser for investigating the speed of the new estimators on an FPGA. This work was supported by the European Research Council (ERC CoG-724489 to J.R.). Open access funding provided by University of Vienna.

## Author contributions

J.R. designed the simulator and the examples. J.R. implemented the MATLAB software. Z.M. implemented the Python software. All authors collaborated to design the estimators. All authors wrote the manuscript.

## Competing interests

The authors declare no competing interests.
