## [Transparent Peer Review file · Nature Communications]

Evaluating MINFLUX experimental performance in silico

Corresponding Author: Professor Jonas Ries

Version 0:

Reviewer comments:

Reviewer #1

(Remarks to the Author)

The authors have carried out all the requested changes.

In particular they addressed the comparison with data taken from experiments, thus corroborating their numerical methods. The new predictions made with the method do not appear ground-breaking, but will be useful to the community working on improving this experimental technique.

I think that the paper deserves publication, although I remain unsure on the suitability for Nature Communications. I believe this is ultimately an editorial decision.

(Remarks on code availability)

Reviewer #2

(Remarks to the Author)

The authors have addressed my previous concerns, added the caveats & notes as suggested and improved the manuscript. However, there are few small corrections (listed below) on the new edits, which are recommended to be made.

- 1) SI Fig3, Left: The labels for each row (aberrations) and column (PSF shape) are missing from the new version. Please add these for better readability of the figure.
- 2) SI Fig3, caption: '.... 2D and 3D (top hat phase mask) donut (vortex phase mask)' to '....2D (vortex phase mask) and 3D (top hat phase mask) donut'
- 3) Fig 2d: Solid red line (with a peak at around $x = 50$) seems to be the newly added curve. Is it for the iterative estimator or for simple estimator? The iterative estimator curve (without consideration of background) looked very different in the previous version of the figure. Did the slight change in background led to this significant change in the curve?
- 4) SI Fig 7: Please include the methods description for the newly included experimental tracks. Was it measured on Abberior MINFLUX? what sequence and parameters were used?
- 5) SI Fig 8, caption: 'The black line is the extra error for very high photons.' According to the main text, the extra error is not dependent on the photon number. Is the black line the extra error due to the fluorophore kinetics, denoted by σ_{fl} in the main text?

(Remarks on code availability)

Reviewer #3

(Remarks to the Author)

The manuscript provides a thorough theoretical foundation for SimuFLUX, featuring rigorous mathematical derivations and

comprehensive modeling of optical systems, fluorophore dynamics, and estimator algorithms. The authors demonstrate a strong command of MINFLUX principles. However, while the theoretical framework is robust, the lack of direct experimental validation substantially limits the work's broader impact.

For example,

1. The PSF models (e.g., vectorial calculations vs. experimental calibrations) and fluorophore kinetics (Supplementary Information Fig. 8) remain untested against physical systems.

2. The proposed estimators (e.g., iterative least squares in Figure 2c) are theoretically sound but lack empirical benchmarking. Claims about FPGA compatibility and real-time performance (Figure 2j–l) would be significantly strengthened by hardware implementation tests or comparisons with existing experimental MINFLUX data.

3. Simulations of biological structures such as nuclear pores (Figure 2f–i) and motor proteins (referencing prior work) are compelling but would benefit from parallel experimental validation.

(Remarks on code availability)

Reviewer #1 (Remarks to the Author):

The authors have carried out all the requested changes.

In particular they addressed the comparison with data taken from experiments, thus corroborating their numerical methods.

The new predictions made with the method do not appear ground-breaking, but will be useful to the community working on improving this experimental technique.

I think that the paper deserves publication, although I remain unsure on the suitability for Nature Communications.

I believe this is ultimately an editorial decision.

We are happy that the reviewer was satisfied with the changes made and thank them again for their suggestions.

Reviewer #2 (Remarks to the Author):

The authors have addressed my previous concerns, added the caveats & notes as suggested and improved the manuscript.

We are happy the reviewer is satisfied with how we addressed their concerns.

However, there are few small corrections (listed below) on the new edits, which are recommended to be made.

1) SI Fig3, Left: The labels for each row (aberrations) and column (PSF shape) are missing from the new version. Please add these for better readability of the figure.

We have added row and column labels.

2) SI Fig3, caption.: '... 2D and 3D (top hat phase mask) donut (vortex phase mask) ...' to '...2D (vortex phase mask) and 3D (top hat phase mask) donut ...'

Updated.

3) Fig 2d: Solid red line (with a peak at around $x = 50$) seems to be the newly added curve. Is it for the iterative estimator or for simple estimator? The iterative estimator curve (without consideration of background) looked very different in the previous version of the figure. Did the slight change in background led to this significant change in the curve?

The new curve (solid red) is for the iterative estimator, in presence of background when the background is not corrected for. It was added because of a reviewer suggestion. None of the other curves changed, we only added this additional curve.

For small x it is similar to the simple estimator, for larger x it outperforms it. In comparison to Figure 2d, which shows the iterative estimator without any background, the only difference is the presence of background that leads to this very different curve.

4) SI Fig 7: Please include the methods description for the newly included experimental tracks. Was it measured on Abberior MINFLUX? what sequence and parameters were used?

We now include a paragraph in the methods on the single-molecule experiments.

5) SI Fig 8, caption: 'The black line is the extra error for very high photons.' According to the main text, the extra error is not dependent on the photon number. Is the black line the extra error due to the fluorophore kinetics, denoted by σ_{fl} in the main text?

Yes, and we have updated the caption.

Reviewer #3 (Remarks to the Author):

The manuscript provides a thorough theoretical foundation for SimuFLUX, featuring rigorous mathematical derivations and comprehensive modeling of optical systems, fluorophore dynamics, and estimator algorithms. The authors demonstrate a strong command of MINFLUX principles. However, while the theoretical framework is robust, the lack of direct experimental validation substantially limits the work's broader impact.

We thank the reviewer for their praise of our MINFLUX theory and their concerns, and do our best to address specific issues below.

For example,

1. The PSF models (e.g., vectorial calculations vs. experimental calibrations) and fluorophore kinetics (Supplementary Information Fig. 8) remain untested against physical systems.

The PSF models are taken from reference 24, Leutenegger, M., Rao, R., Leitgeb, R. A. & Lasser, T. Fast focus field calculations. *Opt. Express* 14, 11277–11291 (2006) and are generally accepted as accurate models. While we agree that the specific implementation of fluorophore kinetics are untested against physical systems, modeling fluorophore kinetics of single molecules using ON and OFF rates is an established practice in the field, particularly for fluorescent proteins (see reference, 40 Lelek, M. et al. Single-molecule localization microscopy. *Nat. Rev. Methods Primer* 1, 39 (2021)). For DNA-PAINT, the kinetics can be exactly modeled (again, reference 40).

2. The proposed estimators (e.g., iterative least squares in Figure 2c) are theoretically sound but lack empirical benchmarking. Claims about FPGA compatibility and real-time performance (Figure 2j–l) would be significantly strengthened by hardware implementation tests or comparisons with existing experimental MINFLUX data.

To ensure we do not mislead readers, we do not make any claims about speed with regards to these estimators in the manuscript. The original Balzarotti paper points out that estimators with mathematical simplicity are easy to implement on an FPGA, and so we aimed for simple equations when developing new estimators.

We performed our own implementation of the conventional estimator on an FPGA, which needs 300 ns for a 1D localization. The iterative estimator is of similar complexity; thus 10 iterations should take approx. 3 μ s.

True comparisons would require reprogramming the FPGA on the Abberior MINFLUX, which is not possible due to intellectual property and warranty concerns.

3. Simulations of biological structures such as nuclear pores (Figure 2f–i) and motor proteins (referencing prior work) are compelling but would benefit from parallel experimental validation.

We agree that comparing our simulator to real-world results would be ideal, and plan to do so in the future. Unfortunately, we do not currently have access to a functional MINFLUX microscope. Since our simulator is in agreement with our previous data and with data published by other groups (<https://www.biorxiv.org/content/10.1101/2025.06.10.658837v1>), we believe the tool is already useful and are sharing it with the community.